

# The impact of landscape evolution on soil physics: Evolution of soil physical and hydraulic properties along two chronosequences of proglacial moraines

Anne Hartmann[1], Markus Weiler[2], and Theresa Blume[1]

[1]GFZ German Research Centre for Geosciences, Section Hydrology, Potsdam, Germany
[2]University of Freiburg, Chair of Hydrology, Freiburg Germany

**Correspondence:** Anne Hartmann (aha@gfz-potsdam.de)

**Abstract.** Soil physical properties highly influence soil hydraulic properties which define the soil hydraulic behavior. Thus, changes within these properties affect water flow paths and the soil water and matter balance. Most often these soil physical properties are assumed to be constant in time and little is known about their natural evolution. Therefore, we studied the evo-
lution of physical and hydraulic soil properties along two soil chronosequences in proglacial forefields in the Central Alps,
5 Switzerland. One soil chronosequence developed on silicate and the other on calcareous parent material. Each soil chronose-
quence consisted of 4 moraines with the ages of 30, 160, 3 000, and 10 000 years at the silicate forefield and 110, 160, 4 900,
and 13 500 years at the calcareous forefield. We investigated bulk density, porosity, the content of clay, silt, sand, and gravel
as well as loss on ignition and hydraulic properties in form of retention curves and hydraulic conductivity curves. Samples
were taken in three depths (10, 30, 50 cm) at six sampling sites at each moraine. Soil physical and hydraulic properties change
10 considerably over the chronosequence. Particle size distribution shows a pronounced reduction in sand content and an increase
in silt and clay content over time at both sites. Bulk density decreases and porosity increases during the first 10 millenia of soil
development. The trend is equally present at both parent materials, but the reduction in sand and increase in silt content was
more pronounced at the calcareous site. The organic matter content increases, which is especially pronounced in the top soil
at the silicate site. With the change in physical soil properties and organic matter content the hydraulic soil properties change
15 from fast draining coarse textured soils to slow draining soils with high water holding capacity, which is also more pronounced
in the top soil at the silicate site. The dataset presented in this paper is available at the online repository of the German Research
Center for Geosciences (GFZ, Hartmann et al. (2020b)). The dataset can be accessed via the link:
http://pmd.gfz-potsdam.de/panmetaworks/review/f46bd4d822a0766a9c0baf356bc7e55644d65d62d7ab71527f5d80c35eed11e5
and will be published with the DOI specified under the link.

## 1 Introduction

Today's landscapes are affected by changes e.g. in form of climate conditions or land use. Insights into the complex and
dynamic interplay between soil development and hydrological, geomorphological and ecological processes in the context of
landscape evolution provide important process understanding which is important for predicting how landscapes will adapt to





changes. Soil has a crucial role in landscape evolution since it influences and is also in turn influenced by vegetation, water, sediment and solute transport. The soil properties are state variables that play an important role within this feedback cycle (van der Meij et al., 2018). The soil physical properties such as bulk density, porosity, and grain size distribution highly influence water flow (flow rates and flow direction), water storage, capacity and drainage [Hu et al. (2008), Lohse and Dietrich

(2005), Reynolds et al. (2002)] as well as root water availability (Hupet et al., 2002). In the course of soil development, these properties change and are influenced by (but in turn also influence) flora, fauna, and water availability. Over time, this interaction also leads to a change in soil water and material transport and their balance (Lohse and Dietrich, 2005). Knowledge of the development of physical and hydraulic soil properties, as well as their codependency, can provide important insights into the changes in hydraulic water balance and water availability during landscape evolution.

Several studies have focused on the alteration of soil biological, chemical and physical properties during soil development by studying soil chronosequences [ Crocker and Major (1955), Egli et al. (2010), Dümig et al. (2011), Vilmundardóttir et al. (2014), D'Amico et al. (2014), Hudek et al. (2017), Musso et al. (2019)]. Especially glacial forefields were proven suitable for this 'space for time approach' as soil develops rapidly on glacial till [e.g. Crocker and Major (1955), Douglass and Bockheim (2006), He and Tang (2008), Dümig et al. (2011), Vilmundardóttir et al. (2014), D'Amico et al. (2014)]. The most commonly

studied soil properties are pH-value, organic carbon, total nitrogen, calcium carbonate, bulk density, and particle size distribution. It was found that in the very first one hundred years of soil development the pH-value decreases fast and that total nitrogen and soil organic carbon increase with the onset of vegetation [Crocker and Major (1955), Vilmundardóttir et al. (2014), Egli et al. (2010)]. The young and poorly sorted soils with no depth dependent property distribution (Crocker and Major, 1955) eventually develop into a layered soil system with vertical gradients in soil properties such as organic matter, color, bulk den-

sity, or particle size distribution. In these geological rather short observation periods (<200 yrs) soils show a high variability in particle size distribution within single age classes, without a specific trend in grain size distribution (Dümig et al., 2011). An extension of the observation period to several thousand years of soil development revealed an accumulation of clay-sized particles with increasing age [Douglass and Bockheim (2006), Dümig et al. (2011)]. Another common finding is also a decrease in bulk density [Crocker and Major (1955), Crocker and Dickson (1957), He and Tang (2008), Vilmundardóttir et al. (2014)]. The

decrease in bulk density is often linked to ongoing vegetation succession, which causes an accumulation of organic matter and the development of a root system. Soil organic matter is also known to have an impact on soil hydrology since it influences soil structure as it contributes to aggregate formation and increases water holding capacity. An increase in organic matter content was also found in a number of soil chronosequence studies [Burga et al. (2010), Douglass and Bockheim (2006), Deuchars et al. (1999), Alexander and Burt (1996)]. Especially in former glacial areas of cool and humid climate, the formation of soils

with a highly organic top layer is favored (Carey et al., 2007). The hydraulic behavior of these organic soils are less intensively studied than the mineral soils. Organic soils for example have a high total porosity (up to 90%) and a low bulk density (Carey et al., 2007). The listed soil chronosequence studies showed that soil texture and soil structure change over time not only due physical and chemical processes, but also due to the influence of vegetation [Morales et al. (2010), Hudek et al. (2017)]. A change in soil structure and texture leads to a change in the soil's behavior which has a direct impact on the surface and sub-

surface water transport.





However, the focus of previous studies was primarily on the estimation of development rates of mainly chemical soil properties and the interaction between soil development and vegetation succession. Only a few studies have looked at how changing physical soil properties and organic matter content affect soil water transport [Lohse and Dietrich (2005), Yoshida and Troch (2016), Hartmann et al. (2020a)], or have focused directly on the development of soil hydraulic properties in form of retention

curves and hydraulic conductivity curves [Crocker and Dickson (1957), Deuchars et al. (1999), Lohse and Dietrich (2005)]. Both curves describe the soil hydraulic behavior and are highly soil specific, since they depend strongly on soil physical properties. The retention curve is the relationship between volumetric soil water content and soil matric potential. The unsaturated hydraulic conductivity curve describes the relationship between unsaturated soil hydraulic conductivity and soil matric potential (or soil water content). While the hydraulic conductivity function gives information on how much water per unit of time

can be transported through the partially filled pore system of the soil matrix at a certain matric potential, the retention curve gives information on how much water is available at a certain matric potential and how much the matric potential will change when a certain amount of water is removed from the soil. Based on these two relationships, the specific soil hydraulic behavior with characteristics such as storage capacity, drainability and the amount of plant-available water can be derived. Additionally, both non-linear relationships are important for the parameterization of physically based soil hydraulic models [Schwen et al.

(2014), Bourgeois et al. (2016)].

Previous investigations of the soil hydraulic properties development are either only based on a few data points, and time periods that are either very long (comparing a 200 and a 4.1 million year old soil) or short (studying soils ranging from less than 50 to 200 years in age) and a small vertical and horizontal spatial resolution. Crocker and Dickson (1957) for example, determined the field capacity of soil samples on the basis of water content measurements after oven drying at 110 °C and after centrifuging

in a standard moisture equivalent centrifuge. Here, an increase in field capacity over the first 200 years of soil development was shown for two glacier forefields in south-eastern Alaska, where the soils developed on glacial till mainly composed by quartz diorite. Lohse and Dietrich (2005) compared in-situ field measurements of water content and matric potential, as well as experimentally derived in-situ unsaturated hydraulic conductivities in the field in two depths of a 300 and a 4.1 million year old site on the Hawaiian Islands. From 300 to 4.1 million years age, the hydraulic characteristics changed significantly with

a shift from rather homogeneous to a layered system with strong differences in the hydraulic characteristics between the soil horizons, which developed from volcanic deposits (Lohse and Dietrich, 2005).

The development of soil hydraulic properties during the first few millennia of soil development has so far not been investigated. We therefore focused on the co-evolution of soil hydraulic properties, soil physical properties and organic matter content during the first 10 millenia of landscape evolution by using soil chronosequences at two glacier forefields. We chose forefields

developed from silicate (Stone glacier forefield) and calcareous rocks (Griessfirn forefield). The study is expected to provide information on how strong the evolution of soil physical properties affects the development of soil hydraulic properties and to give insights about the changes to be expected in non-stationary landscape systems. The comparison of two sites is expected to highlight how the parent material influences the development. A detailed investigation of hydrologic flow path evolution on these same moraines can be found in Hartmann et al. (2020a).

The data set can be useful to improve predictions on hydrological processes during landscape development using soil and





landscape evolution models (SLEMs). van der Meij et al. (2018), for example, proposed the incorporation of measured soil hydraulic properties from chronosequence studies in the calibration of SLEMs to account for the long term evolution of soil hydraulic properties. This is important to improve the feedback modeling between soil structure and soil hydrologic processes as well as for an improved process reflection of the interaction of pedogenic, geomorphic and hydologic processes. Our data

set is also suitable for the derivation or verification of pedotransfer functions for alpine soils. Pedotransfer functions are a less time-consuming and cost-effective method for determining the soil hydraulic properties (Vereecken et al., 2010) and are also used in SLEMs (van der Meij et al., 2018). They are designed to translate easy-to-measure soil properties such as organic carbon content, bulk density, grain size distribution or porosity [Wang et al. (2009), Schaap et al. (2001)] into soil hydraulic properties. The data set makes it possible to derive and test pedotransfer functions for both study sites in order to find out

whether site properties such as parent material have an additional influence on the validity of the pedotransfer function. Further soil physical and hydraulic properties from chronosequence studies can be helpful to derive information of water and nutrient availability, which can be important for other chronosequence studies related to abundance, diversity and function of microbial life in initial soils as well as for studies of vegetation succession.

## 2   Material and methods

### 15  2.1   Study sites

We investigated how soil structure and soil hydraulic behavior change through time by using a soil chronosequence at two glacier forefields. The two study sites differ in their parent material. The selected proglacial moraines at the Stone Glacier forefield developed from silicate parent material (S-PM) and the moraines at the Griessfirn forefield from calcareous rocks (C-PM). The parent material is one of the five main factors of soil formation (next to climate, biota, topography, and time). The

comparison of the two parent materials is expected to provide information on how under assumed equal climate conditions this site characteristic influences the development of structure and soil hydraulic behavior. It is already known that soils developed on calcareous material are richer in organic carbon and clay particles (Jenny, 1941), but little is known how strong these differences are throughout the course of soil development and how much they influence the soil hydraulic behavior.

### 2.1.1   Silicate parent material

The study area of the proglacial forefield developed on silicate parent material was formed by the retreat of the Stone Glacier and is located in the Central Swiss Alps, south of the Sustenpass in the Urner Alps (appr. 47° 43'N, 8° 25'E). Its elevation ranges from 1900-2100 m a.s.l. The area lies in the polymetamorphic "Erstfelder" gneiss-zone, which is part of the Aar-massif (Blass et al., 2003). The geology is defined by metamorphosed pre-Mesozoic, metagranitoids, gneisses, and amphibolites (Heikkinen and Fogelberg (1980), Schimmelpfennig et al. (2014)), thus the material is mainly acidic and rich in silicate.

The closest official weather station is located 18 km away at Grimsel Hospiz (46° 34'N, 8° 19'E) at an elevation of 1980 m a.s.l. The recorded annual mean temperature is 1.9 °C and the annual precipitation is 1856 mm (1981-2010) (Schweizerische



Eidgenossenschaft, 2016). The moraines of the Stone Glacier were exposed due to its retreat to the south. Four moraines were selected for this study (see Fig. 1). Schimmelpfennig et al. (2014) conducted a detailed dating study of the Stone Glacier moraines, based on high-sensitivity beryllium-10 moraine dating and found that the ages of the four moraines range between 160 to 10 000 years. The age of the youngest moraine was dated as 30 years based on maps and aerial photos.

The vegetation cover differs significantly among the four age classes and was mapped in summer 2017 (Maier et al., 2019). The moraines are occasionally grazed by cows and sheep during the summer months, which we prevented during our study by the installation of fences. Whereas the vegetation cover at the oldest moraine was dominated by a variety of prostrate shrubs, small trees and several grasses, the 3 000-year-old moraine has mainly a grassland cover with fern, mosses, sedges and forbs. The 160-year-old moraine was dominated by grasses, lichen, forbs, and shrubs. The vegetation cover of the youngest moraine
was sparse with mainly grass, moss, forbs, and a few shrubs.

### 2.1.2   Calcareous parent material

The study area of the proglacial forefield developed on calcareous parent material was formed by the retreat of the Griessfirn and is located between 2030-2200 m a.s.l. in the Central Swiss Alps (appr. 46° 85'N, 8° 82'E). The geology is defined by limestone (Frey, 1965), thus the material is mainly calcareous. A more detailed description of the geological composition is
provided by Musso et al. (2019). The closest official weather station located at a similar elevation (2106 m a.s.l.) is 48 km away at Pilatus Mountain (46° 98'N, 8° 25'E). The recorded annual mean temperature is 1.8 °C and the annual precipitation is 1752 mm (1981-2010) (MeteoSwiss, 2020).

The four selected moraines were dated by Musso et al. (2019) based on historical maps and the radiocarbon method. The youngest moraine is 110 years old and is located at 2200 m a.s.l. The three other moraines are 160, 4 900, and 13 500 years
old and located at an elevation of roughly 2030 m a.s.l. (see Fig. 1). The two oldest moraines were densely covered with grass, dwarf shrubs and sedge. The vegetation coverage of the two younger moraines was sparse with patches of grass and forbs at the 160-year-old moraine and patches of mostly mosses and lichens at the 110-year-old moraine.



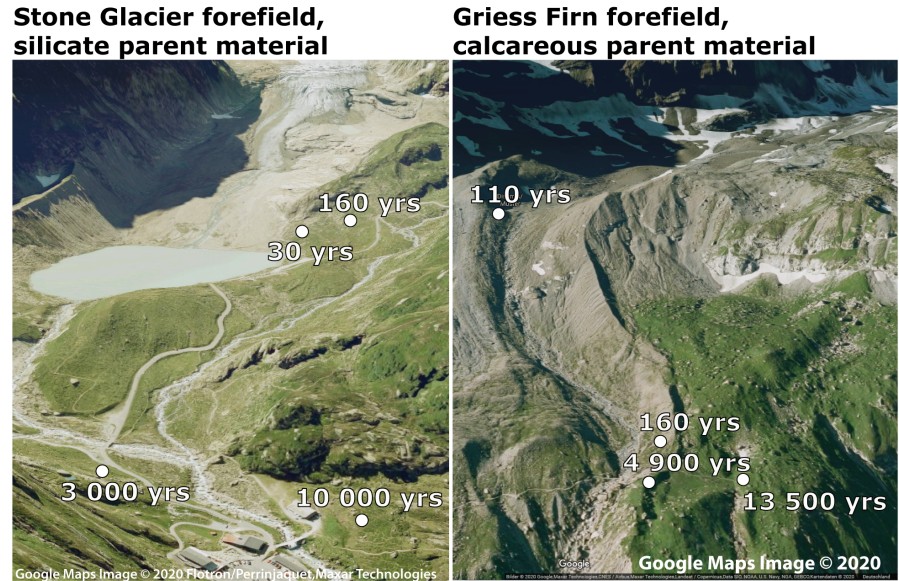

**Figure 1.** Glacier forefield and location of the four selected moraines of the silicate parent material (S-PM, left (Google, n.d.a)) and the calcareous parent material (C-PM, right (Google, n.d.b)).

## 2.2 Soil sampling and laboratory analysis

### Soil sampling

Soils samples were taken during August and September of 2018 at the silicate site and during August and September 2019 at the calcareous site. Three sampling sites were chosen per moraine to capture three complexity levels (low, medium, high) of the vegetation coverage (Musso et al., 2019). Table 1 provides a detailed overview of the sampling scheme at each sampling site for both parent materials. At each sampling site, replicate samples were taken 3-4 m apart to account for spatial variability. The two sampling locations per sampling site are denoted as Location 1 and Location 2 in Table 1. For grain size analysis, at each sampling site two disturbed soil samples (one per sampling location) were taken at 10, 30, and 50 cm depth.

**Table 1.** Overview of the sampling scheme at each sampling site for both parent materials. The two sampling locations per sampling site are denoted as Location 1 and Location 2. Locations 1 and 2 were 3-4 m apart. The volume of the sample rings is given in the respective headers and the corresponding numbers of samples are provided for each soil depth.

| | *Silicate parent material* | | | | | | *Calcareous parent material* | | | | | |
| --- | --- | --- | --- | --- | --- | --- | --- | --- | --- | --- | --- | --- |
| | *Location 1* | | | *Location 2* | | | *Location 1* | | | *Location 2* | | |
| | $100\,cm^3$ | $250\,cm^3$ | *disturbed* | $100\,cm^3$ | $250\,cm^3$ | *disturbed* | $100\,cm^3$ | $250\,cm^3$ | *disturbed* | $100\,cm^3$ | $250\,cm^3$ | *disturbed* |
| *Depth* [*cm*] | | | | | | | | | | | | |
| 10 | 1 | 2 | 1 | - | 2 | 1 | - | 2 | 1 | - | 2 | 1 |
| 30 | 1 | 2 | 1 | - | 2 | 1 | - | 2 | 1 | 1 | 1 | 1 |
| 50 | 2 | 1 | 1 | 1 | - | 1 | 1 | 1 | 1 | 2 | - | 1 |





For the determination of porosity, bulk density, loss on ignition and the derivation of the soil hydraulic properties undisturbed soil samples were taken with steel sampling rings which preserve the natural soil structure. At each sampling site and sampling location at the S-PM, two 250 cm$^3$ undisturbed soil samples were taken at a depth of 10 and 30 cm and one 100 cm$^3$ sample was taken at a depth of 50 cm. Additionally at sampling Location 1, one undisturbed 100 cm$^3$ soil sample was taken at a

depth of 10 and 30 cm and one 250 cm$^3$ and two 100 cm$^3$ soil samples were taken at a depth of 50 cm. This sampling scheme provides 15 undisturbed soil samples at the depths of 10 and 30 cm and 12 samples at a depth of 50 cm per moraine. All 168 samples were used for the determination of porosity and bulk density. Due to the high stone content at the S-PM forefield, a few sampling rings were damaged and could not be used the following year. Therefore, the sampling scheme at the C-PM forefield had to be adapted to a reduced number of samples (see tab. 1). At the C-PM forefield and Location 1 at each sampling site, two

250 cm$^3$ undisturbed soil samples were taken at a depth of 10 cm and 30 cm and one 250 cm$^3$ and one 100 cm$^3$ sample were taken in 50 cm. At Location 2, two 250 cm$^3$ samples were taken in 10 cm and one 250 cm$^3$ and one 100 cm$^3$ sample in 30 cm as well as two 100 cm$^3$ samples in 50 cm. This sampling scheme provides 12 undisturbed soil samples at the depths of 10, 30 and 50 cm per moraine. All 144 samples were used for the determination of porosity and bulk density.

**Laboratory analysis**

The laboratory analysis was carried out between October 2018 and June 2019 for the S-PM samples and between October 2019 and January 2020 for the C-PM samples. For the grain size analysis, we used a combination of dry sieving (particles > 0.063 mm) and sedimentation analysis (particles < 0.063 mm) with the hydrometer method. Particles between 2 mm and 0.063 mm were classified as sand, between 0.063 mm and 0.002 mm as silt and < 0.002 mm as clay. Organic matter removal was only possible by floating off the lighter fractions prior to particle size analysis and in case of the particle fraction >0.063 mm

by combustion at 550 °C prior to the dry sieving. Particle size fractions were calculated as weight percentages of the fine earth (< 2 mm), thus excluding gravel and stones to avoid that single larger stones shift or dominate the distribution. The gravel and stone fraction (particles > 2 mm) was calculated separately as a weight percentage of the entire soil sample. The porosity was determined by using the water saturation method and weighing the samples at saturation and after drying at 105 °C. The loss on ignition was determined by drying sub-samples (4-6 g) of 131 samples of the S-PM forefield and 144 samples of the C-PM

forefield for at least 24 hours at 105 °C and then at 550 °C. The ignition loss is then calculated by relating the weight loss after drying at 550 °C to the sample weight after drying at 105 °C.

15 undisturbed 250 cm$^3$ samples per moraine (6 samples at both 10 and 30 cm depth and 3 at 50 cm depth) were used for the analysis of soil hydraulic properties in form of retention curve and hydraulic conductivity curve. The soil hydraulic properties of the 120 soil samples were measured based on the experimental evaporation method [Schindler (1980), Schindler and Müller

30 (2006)]. This laboratory based method allows the simultaneous determination of retention curve and hydraulic conductivity curve. The water saturated soil samples are dried evenly and slowly by evaporation. During this process, the weight of the soil sample and the matric potential in two heights in the soil core are measured. To conduct the experiment we used the ku-pF MP10 (Umwelt-Geräte-Technik GmbH, Germany). The device allows to conduct the experiment simultaneously on ten soil samples at a time, with each cycle taking two to three weeks. The device holds ten soil samples on a rotating appliance



and automatically measures the weight of each sample. The interval between individual sample measurements was set to one minute, so that each sample was measured every ten minutes. For the measurement of the matric potential two tensiometers were installed in two different heights in the soil core. At each weight measurement, the device also records the tensiometer readings. The data analysis was carried out according to Peters and Durner (2008). Based on the weights the soil water content

5  can be derived and is set in relation with the average measured matric potential to provide the retention curve. Based on the measured reduction in water content a flow rate can be determined and the hydraulic gradient can be derived based on the measured matric potentials. The combination of flow rate and gradient allows the determination of the unsaturated hydraulic conductivity for all measured water contents.



# 3 Data set of soil physical properties and their change through the millenia

## 3.1 Bulk density and porosity

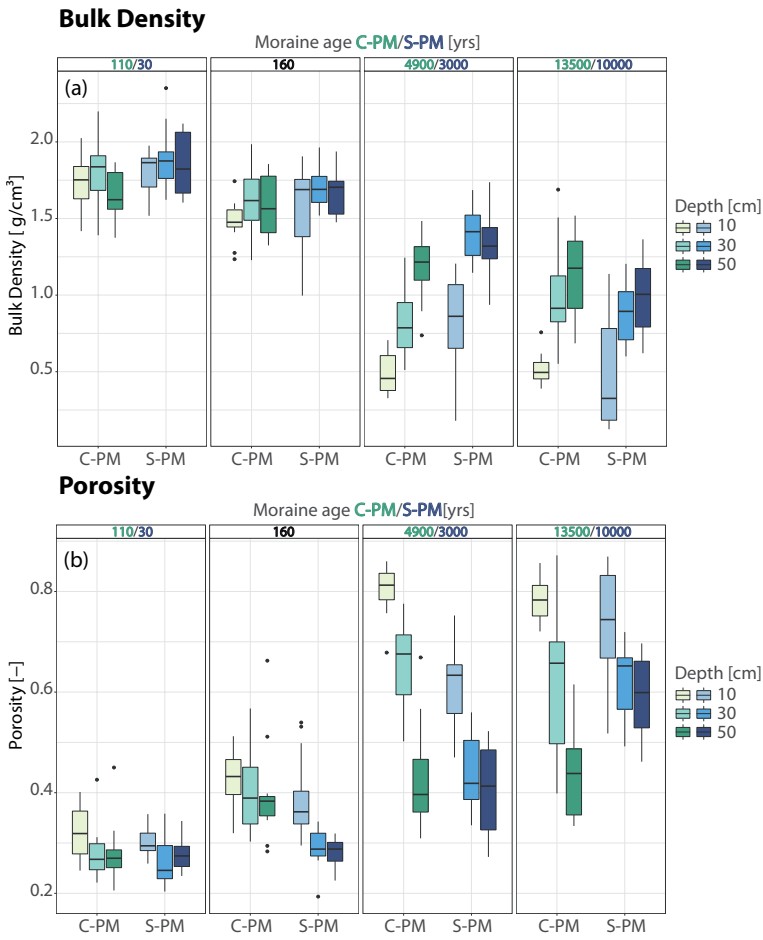

**Figure 2.** Development of bulk density and porosity in 10, 30, and 50 cm depth over 10 millenia on silicate (S-PM, shown in blue color scale) and calcareous (C-PM, shown in green color scale) parent material.

The obtained data sets of bulk density and porosity show a clear trend over the millenia in both properties at both soil chronose-quences (see Fig. 2). At the C-PM forefield, the bulk density decreases along the chronosequence in all soil depths (see Fig. 2a).

5  The decrease is most pronounced in the top layer and is weaker towards deeper soil depths. With increasing age an ongoing differentiation in bulk density along the soil profile is observed. At the youngest moraine of 110 years, the bulk density ranges mainly between median values of 1.6 g/cm$^3$ and 1.8 g/cm$^3$ with slightly higher values in 30 cm. At the 160-year-old moraine, the bulk density in all depths varies in median values between 1.5 and 1.6 g/cm$^3$. The 4 900 year-old moraine has significantly





lower bulk densities, with the lowest values in the top layer (median value in the top layer 0.46 g/cm$^3$). The bulk density increases with depth and varies around 0.78 g/cm$^3$ in 30 cm and around 1.2 g/cm$^3$ in 50 cm. The oldest moraine of 13 500 years does not show major differences in bulk density from the 4 900-year-old moraine. The ranges of the values differ but the median values are quite similar.

At the youngest moraine at the S-PM forefield, the interquartile range (IQR) at all three depths overlap, the bulk density values vary mainly between 1.7 and 1.9 g/cm$^3$. The bulk density at the 160-year-old moraine is lower and ranges mainly between 1.4 (lower end of IQR at 10 cm) and ~1.75 g/cm$^3$ (upper end of IQR at all three depths). At this age class, the uppermost layer already shows the tendency to have a lower bulk density than the deeper soil. This is even more pronounced at the 3 000 year old moraine. The bulk density in 30 and 50 cm depth at this moraine ranges between median values of 1.3 (at 50 cm) and 1.4

g/cm$^3$ (at 30 cm), whereas the bulk density in 10 cm mainly ranges in IQR between 0.65 and 1.07 g/cm$^3$. At the 10 000-year-old moraine the bulk density is the lowest and varies in IQR between 0.2 and 0.78 in the uppermost layer. The IQR at 30 and 50 cm overlap strongly. Here, the bulk density varies between 0.7 (lower end of IQR at 30 cm) and 1.17 (upper end of IQR at 50 cm).

A comparison of the 160-year-old moraine at both locations shows that the bulk density across the soil profiles are similar at

both locations, but are in general lower at the C-PM. At the second oldest moraines the bulk density at C-PM is also lower in all three depths compared to S-PM. However, this relation is reversed at the oldest moraine. The bulk density in 10 and 50 cm depth at S-PM is significantly lower, even though the moraine is younger than the corresponding moraine at C-PM.

The porosity shows an increase along the age groups and an ongoing differentiation across the soil profile at both chronosequences (Fig. 2b). At C-PM the porosity at the youngest moraine is in a similar range along the soil profile, except for the top

layer, which shows higher porosity values with a median at 0.32 (30-50 cm median value: ~0.27). At the 160-year-old moraine, the values are already slightly higher and vary around median values of 0.38 and 0.43. The 4 900-year-old moraine shows a strong increase in porosity and a clear differentiation between the individual depths. The porosity in the topsoil is highest with a median value slightly above 0.81. In the layers below, on the other hand, the porosity is lower (values vary around median values of 0.67 and 0.39 in 30 and 50 cm depth, respectively). Analogous to the development of bulk density, there is no striking

difference between the two oldest moraines, the porosity values have a similar range.

At the S-PM forefield the porosity evolution shows similar tendencies. At the 30-year-old moraine the porosity in 30 to 50 cm depth ranges between 0.24 and 0.29. The porosity in 10 cm is slightly higher with values in an IQR from 0.28 and 0.32. A differentiation between the soil depths is, however, already visible at the 160-year-old moraine, where the IQR of porosity values in the uppermost layer ranges from 0.34 to 0.4. The porosity in 30 and 50 cm depth, however, is not noticeably different

from the same depths at the youngest moraine. Compared to the two youngest moraines the porosity in the uppermost layer at the 3 000-year-old moraine is distinctly higher with values of the IQR mainly ranging between 0.55 and 0.65. The porosity in 30 and 50 cm is also higher and varies over a broader range compared to the younger moraines. The IQRs of the two depth are overlapping. The porosity values range mainly between 0.33 and 0.5 with a median value at 0.41 at both depths. After 10 000 years of soil development the porosity reached its highest values ranging in an IQR from 0.67 to 0.83 in the uppermost soil

layer and in an again overlapping IQR at 30 and 50 cm depth from mainly 0.5 to 0.66. In contrast to the 3 000-year-old moraine





the median values differ in 30 and 50 cm depth (0.65 at 30 cm and 0.6 at 50 cm).

A comparison of the two 160-year-old moraines at the two sites shows that the porosity at the C-PM is higher in all three soil depths. However, while there was a strong change in porosity at all depths at S-PM between 3 000 and 10 000 years of soil development, there was no clear difference between the two oldest age groups at C-PM.

5  The evolution of bulk density and porosity does not only reveal a constant decrease in bulk density and increase in porosity, but also shows a progressive differentiation of these values between the soil layers. Additionally, an increase in the range of values is also noticeable. Thus, with increasing age not only the vertical, but also the lateral variability of bulk density and porosity increases.

### 3.2  Soil texture

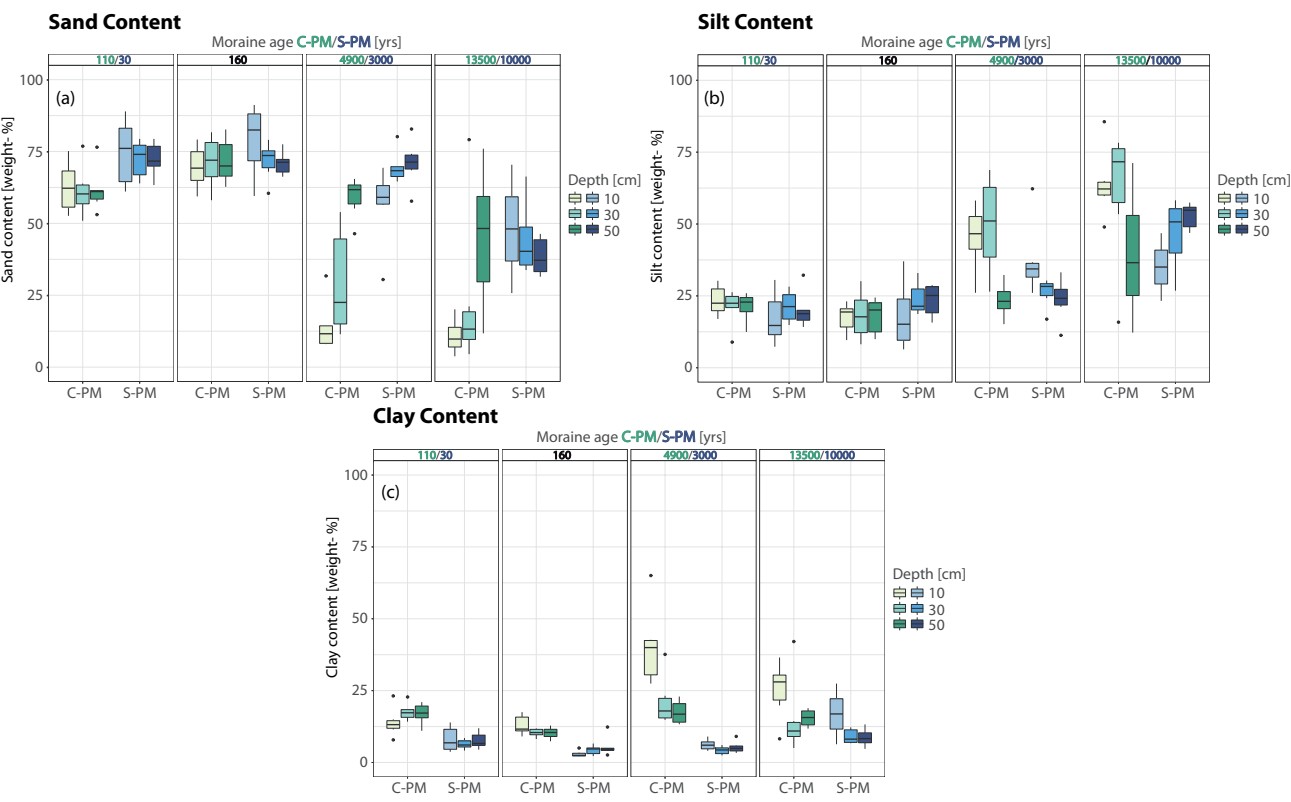

**Figure 3.** Development of sand-, silt-, clay-content in 10, 30, and 50 cm over 10 millenia on silicate (S-PM, shown in blue color scale) and calcareous (C-PM, shown in green color scale) parent material.

10  The development of the grain size distribution over the millenia shows a distinct reduction in the sand fraction in all three depths at both chronosequences (Fig. 3a). At C-PM, the fraction of sand at the youngest moraine with over 50 weight-% accounts for the largest share of all grain sizes and is also relatively homogeneous with depth. The fraction of sand at the



160-year-old moraine is slightly larger compared to the youngest moraine. With increasing age, there is a significant reduction in the fraction of sand, especially in the upper layers. The sand content at the surface is reduced to ~10 weight-% at the two oldest moraines, whereas the sand content in 50 cm is less affected. However, in 30 cm at the 4 900-year-old moraine and in 50 cm at the 13 500-old-moraine the sand fraction varies over a broad range (15-45 weight-% and 30-60 weight-%, respectively).

At S-PM, the fraction of sand at the youngest moraine is relatively homogeneous across the soil profile. At the 160-year old moraine, the fraction at the topsoil is slightly higher than at the youngest moraine. However, a differentiation with depth can already been seen with lower values in deeper layers. The reduction in sand content continues with increasing age, whereas at the 3 000 year-old moraine the distribution with depth is reversed with the topsoil having a lower sand fraction than the deeper layers. At the 10 000 year-old-moraine, the profile distribution reverses again with the topsoil having now the highest values.

In general, the fraction of sand in the individual depths and age classes is higher at the S-PM chronosequence compared to the C-PM chronosequence.

It has to be taken into account that the particle size analysis of the samples at a depth of 10 cm in the 10 000-year-old moraine at S-PM could only be carried out on two samples since the organic matter content of the other samples was too high. Since a complete removal of the organic matter content cannot be guaranteed, the two samples are subject to strong uncertainties.

These uncertainties also apply to the other samples from the oldest moraines at both chronosequences that have a high organic matter content.

For the silt fraction, an increase in all depths over the chronosequence can be seen at both forefields. This is more pronounced at C-PM than at S-PM (Fig. 3b). At C-PM, the silt fraction at the 110 and 160-year-old moraines is mostly lower than 25 weight-% and homogeneous across the profile, but with slightly higher values at the youngest moraine. With increasing age,

the silt fraction increases strongly, especially in 10 and 30 cm. After 13 500 years the silt fraction is at its highest in 30 cm (median: ~ 72 weight-%), even higher than in the topsoil (median: 62 weight-%) and the lowest in 50 cm (median: 37 weight-%).

At S-PM, the silt fraction in the top soil at the youngest moraine (median: <20 weight-%) is slightly lower than in the deeper soil (median ~20 weight-%). The difference can be seen more clearly at the 160-year-old moraine, where the silt fraction in

all depths is still mainly below 25 weight-%. After 3 000 years of soil development the silt content in 10 and 30 cm is higher compared to the 160-year-old moraine, but the distribution reverses revealing a decrease in silt content with depth. The silt fraction in 50 cm is in the same range as at the 160-year-old moraine (median: ~25 weight-%). Whereas in the top layers the silt content increased to values ranging in the IQR from 31-35 weight-%. After 10 000 years the depth distribution is reversed again. Compared to the 3 000-years old moraine the silt fraction in the topsoil is still in the same range, whereas the fraction in

30 and 50 cm increased to median values equal and higher 50 weight-%. Among S-PM and C-PM, the depth distribution of the silt content at the oldest moraine differs significantly. Whereas at S-PM the silt content increases with depth, the silt content at C-PM is highest in 30 cm (median value at 72 weight-%) and lowest in 50 cm (median: ~30 weight-%).

The clay content increases with age at both chronosequences (see Fig 3c). At the youngest moraine of both chronosequences the clay fraction is fairly homogeneous across the soil profile. At C-PM the topsoil is having a slightly lower clay fraction

(median: ~13 weight-%) than the deeper layers (median ~17 weight-%), but the values are in general slightly higher than





at S-PM (median in all depths < 10 weight-%). At the 160-year-old-moraine, the clay fraction at both chronosequences are comparatively lower than at the youngest age class, but relatively homogeneous throughout the soil profile. The clay fractions at C-PM (median at all depths: ∼11 weight-%) are higher than at S-PM (median at all depths: < 10 weight-%). At the second oldest moraine of the C-PM chronosequence the clay content increases in all depths, which is most pronounced in the top

5    layer (median value around 40 weight-%). The oldest moraine, however, shows lower values in 10 and 50 cm compared to the second oldest moraine. The clay fraction in the topsoil at 13 500 years (∼28 weight-%) is still higher than in 50 cm, where at 30 cm the clay fraction is the lowest (median ∼10 weight-%). At S-PM only the topsoil shows a slightly increase in clay fraction from 160 to 3 000 years of soil development. While at the oldest moraine clay fraction increases in all depths, with the top soil having the highest fraction (median value ∼17 weight-%). In comparison, the clay content at all age classes is higher

10   at C-PM than at S-PM. A first investigation of the effect of the soil texture at the S-PM forefield on near-surface hydrology can be found in Maier et al. (2019).



Earth System **Science** Data
Open Access · Discussions



**Figure 4.** Ternary diagram for soil type classification using the USDA Textural Soil Classification. Each plot shows a single age class and compares the grain size distributions of both parent materials and all 3 depths.

Incorporated into a ternary diagram (Fig. 4) for soil type classification using the USDA Textural Soil Classification (Hamilton and Ferry, 2018), the grain size evolution reveals a clear shift in soil types at both chronosequences throughout the millenia. The soil types at the two youngest moraines at both chronosequences mainly vary between Loamy Sand and Sandy Loam. Soil types at the 3 000 (S-PM) and 4 900-year (C-PM) old moraines differ from each other (Fig. 4c). Whereas at S-PM Sandy Loam



is still the prevailing soil type, the soil types at C-PM shifted to Silty Clay and Silty Clay Loam in the top soil and mainly Loam and Silt Loam in 30 cm. At 50 cm Sandy Loam is still the main soil type. At the oldest moraine the soil types at S-PM also shift to Loam and Silt Loam (Fig. 4d). At C-PM the topsoil at the oldest moraine is still mainly a combination of Silt, Clay, and Loam. In 30 and 50 cm depth, however, next to Loam and Silt dominated soil also Sandy Loam is frequently present.

5   The gravel and stone fraction (see Fig. 5) here included only stones with a diameter between 2 and ∼100 mm. At both parent materials the gravel and stone fraction decreases with soil age. The decrease is most pronounced in the top soil and at the oldest moraine in the C-PM forefield (Fig. 5(b)).

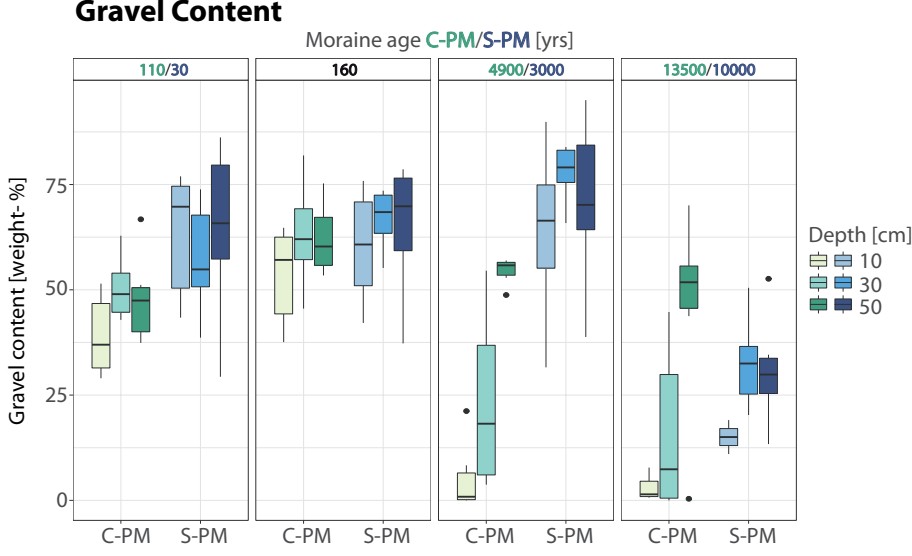

**Figure 5.** Development of gravel/stone content in 10, 30, and 50 cm over 10 millenia on silicate (S-PM, shown in blue color scale) and calcareous (C-PM, shown in green color scale) parent material.

Hartmann et al. (2020a) provides additional information on the evolution of stone content with depth for the S-PM forefield derived by image analysis of soil profile walls with a vertical extent up to 1 m.



### 3.3 Loss on ignition

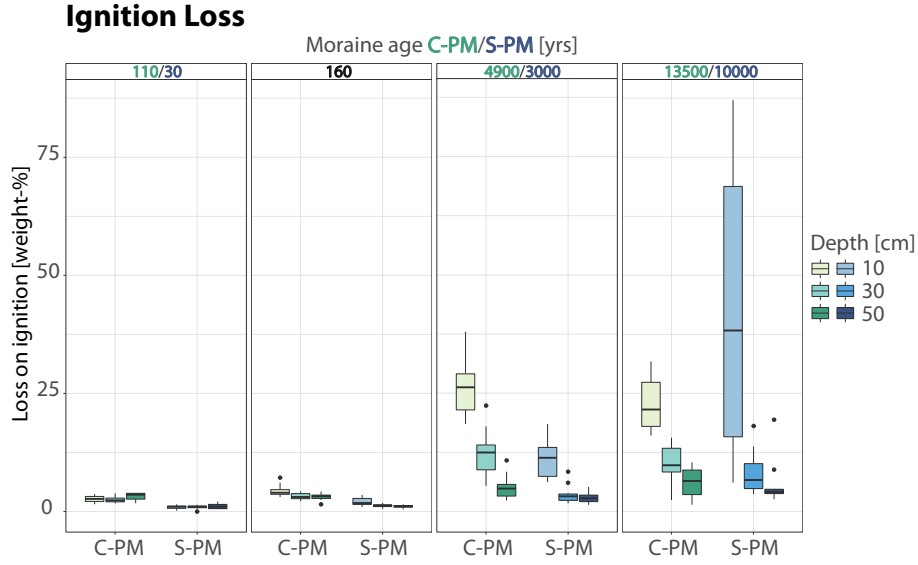

**Figure 6.** Development of loss on ignition in 10, 30, and 50 cm over 10 millenia on silicate (S-PM, shown in blue color scale) and calcareous (C-PM, shown in green color scale) parent material.

The loss on ignition is a measure of the organic substance in the soil and describes the proportion of the organic substance that was oxidized during annealing for 24 hours at 550 °C. The organic substance is a heterogeneous mixture of faunal and floral substances.

5    Both chronosequences show a significant increase in organic matter throughout the first 10 millenia of soil development, which is most pronounced in the upper soil layer (see Fig. 6). For both chronosequences at the two youngest moraines the organic matter content is still very low, with C-PM showing slightly higher values (< 2 weight-% at S-PM and 2-4 weight-% at C-PM). At these two age classes the organic matter content is homogeneously distributed over the profile, with a slight tendency to higher values in the topsoil at the 160-year-old moraine. At the medium age moraines of both chronosequences (3 000 and 4 900

10    yrs) a significant increase in the organic matter content in the surface layer can be observed (median: 11 weight-% at S-PM and up to 26 weight-% at C-PM). There is also an increase at greater depths, which is more pronounced at C-PM. At the oldest moraine at S-PM the trend of increasing organic matter continues in all three depths. Here, the organic content in the topsoil makes up to two-thirds of the soil material. However, the organic matter content varies strongly with a minimum of 6 and a maximum 87 weight-%. At greater depths, the organic matter content also increases compared to the 3 000 year old soil, but

15    remains below 20 weight-%. The organic matter content decreases with increasing soil depth. At C-PM the organic matter content at the top layer of the oldest moraine is slightly lower compared to the second oldest moraine. In general the organic matter content in all three soil depths does not differ considerably between the two age classes.





## 4 Data set of soil hydraulic properties and their change through the millenia

### 4.1 Retention curves

The retention curve, the relationship between volumetric soil water content and matric potential, is an important individual characteristic of soils, and depends strongly on soil physical and biological soil properties. The retention curves show a clear

5    change over the millennia at both forefields (Fig. 7).

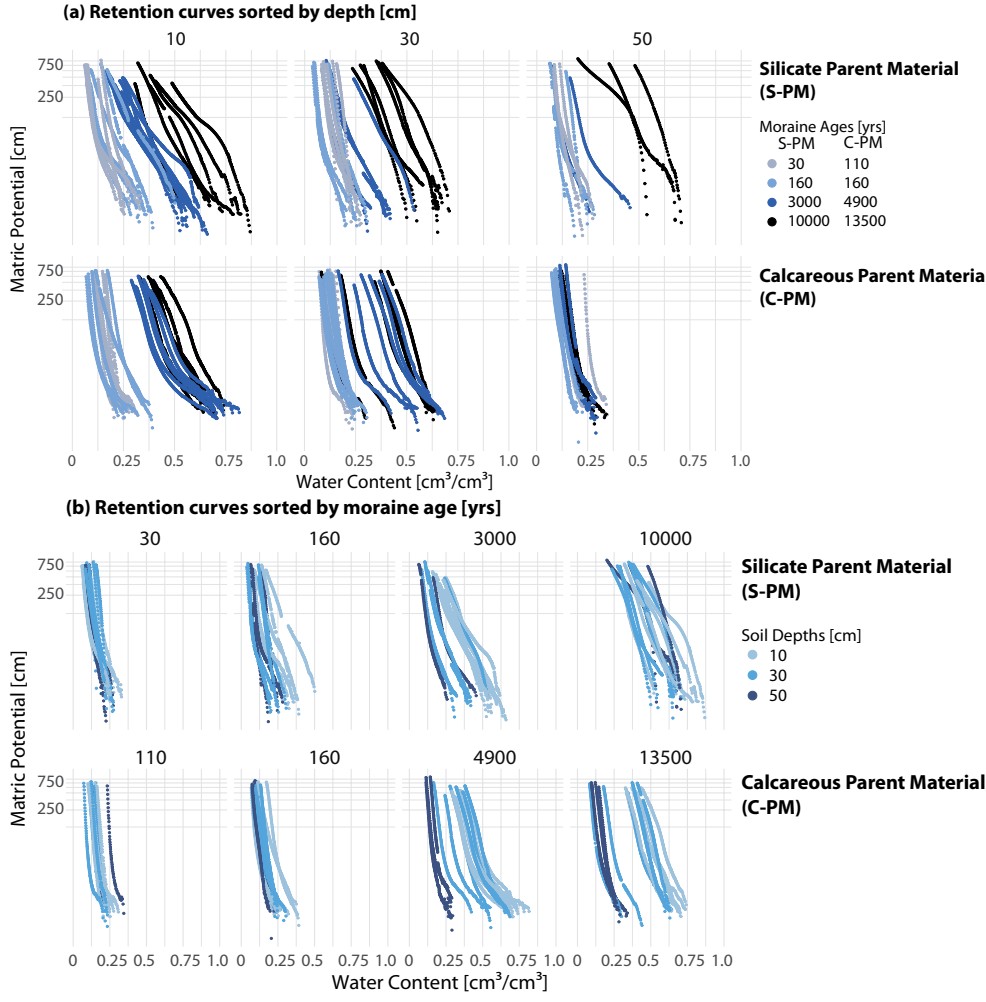

**Figure 7.** Development of retention curves in 10, 30, and 50 cm over 10 millenia on silicate (S-PM) and calcareous (C-PM) parent material sorted by depth (a) and moraine age (b).

At both chronosequences the lower end of the retention curves (nearly saturated conditions) show a clear shift to higher water contents with increasing moraine age, which is strongly coupled to the increase in porosity (equal to saturated water content)





(see Fig. 7a). This trend is most pronounced in the top layer and decreases with soil depth. The air entry value, indicated by the first change in slope close to saturated conditions, shifts to higher matric potential, which is particularly pronounced in the organic layer at S-PM. At C-PM, however, this shift is not very pronounced and especially in 50 cm depth, the retention curves are all similar. However, at both chronosequences the slope of the curve above the air entry value shows the tendency to decline

5    with increasing age. The variability in retention curves with depth also increases with age at both chronosequences (see Fig. 7b). After 3 000 years at S-PM and 4 900 years at C-PM, the retention curves of the three depths show clear differences, while at the youngest moraines of both chronosequences the retention curves of the three depths are similar indicating a homogeneous soil profile. For the older moraines, the increase in saturated water content and air entry value, as well as the decrease in slope, are most pronounced in the uppermost layer, this becomes less pronounced with increasing depth.

10   ## 4.2   Hydraulic conductivity curves

The unsaturated hydraulic conductivity curve is another important flow defining soil characteristic. It describes the relation between the unsaturated soil hydraulic conductivity and soil matrix potential (or soil water content, respectively). The hydraulic conductivity curves also change over the millenia (Fig. 8a).

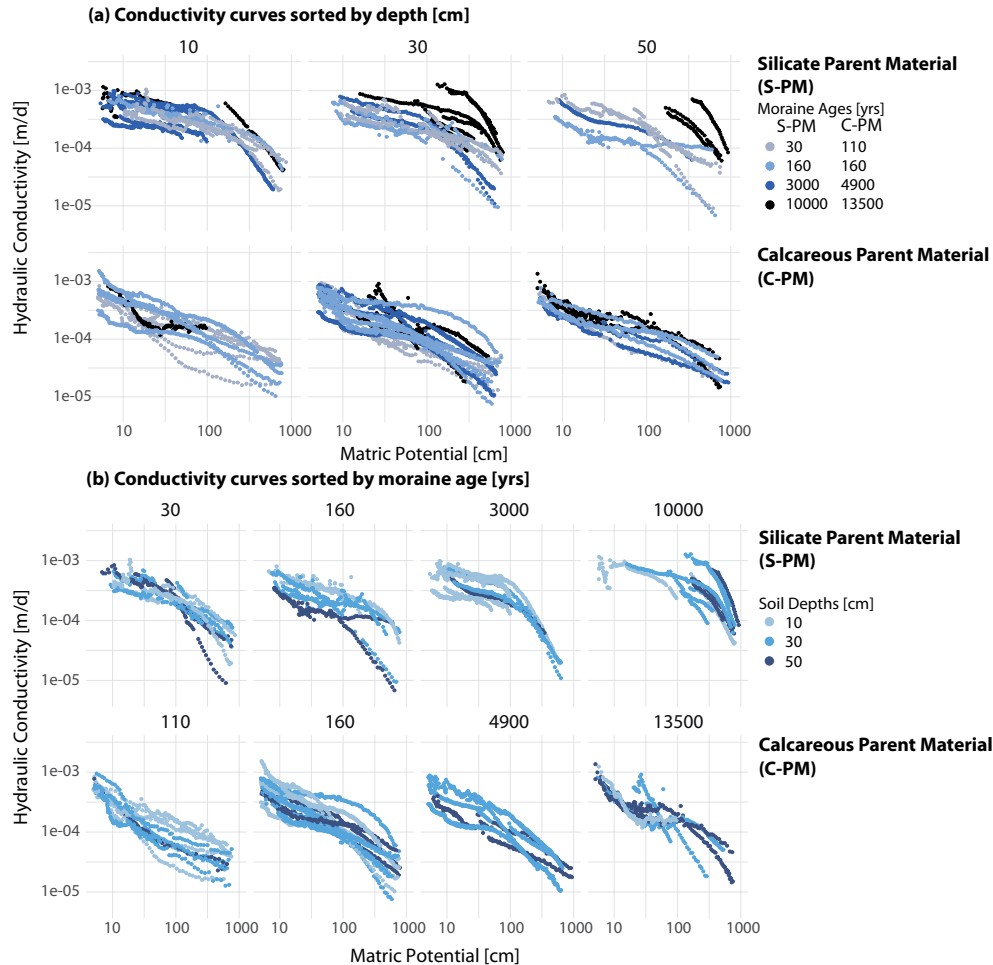

**Figure 8.** Development of hydraulic conductivity curves in 10, 30, and 50 cm over 10 millenia on silicate (S-PM) and calcareous (C-PM) parent material sorted by depth (a) and moraine age (b).

    Only 32 at S-PM and 41 at C-PM of the 60 soil samples that were analyzed by the method according to Schindler (1980) could be analyzed for the hydraulic conductivity curve according to the method by Peters and Durner (2008). Seven of the 28 excluded soil samples at S-PM and two at C-PM could not be used because the installation of a second tensiometer into the sample was prevented by stones. The other excluded samples could not be evaluated, since during the evaporation experiment,
5  the upper part of the sample dried out much faster than the lower part. This leads to an ever-increasing difference between the two measured soil matric potentials during the course of the measurements. An approximately linear profile of the soil matric potential within the sample can thus no longer be assumed. This is usually typical of large-pored soils, such as sandy soils. With the drying of the topsoil the unsaturated hydraulic conductivity decreases rapidly and the water transport from bottom to top is slowed down considerably. To prevent this as far as possible, the ambient conditions were chosen to keep the evaporation





demand as low as possible. The excluded samples belonged to all age classes, but most of them to the top soil of the oldest moraines, where the organic matter content is very high (Fig. 6).

Despite the thin data basis, a trend is visible at both chronosequences that with increasing moraine age the heterogeneity in the conductivity curves increases (Fig. 8). At the S-PM forefield the conductivity curves in 10 cm show a sharp drop in

conductivity, except for the young moraine. Here, the reduction of the conductivity reduces only slowly to a certain point, at which the decrease continues faster. The conductivity curves of C-PM in 10 cm cannot be evaluated in this context, since the curves of the top soil at the two oldest moraines could not be evaluated. However, the curves of the youngest moraine show a similar modest decrease in conductivity at higher matric potentials as at the young moraines at the S-PM forefield. The curves of the 160-year-old moraine at C-PM already show a sharper decrease in conductivity.

At 30 cm depth at S-PM, almost all curves show a slow reduction in conductivity. The curves of the 3 000-year-old moraine are an exception here and show steeper curves. The curves in 30 cm at C-PM are very heterogeneous. The curves of the young moraines still show a rather slow decrease in conductivity, whereas the curves at the older moraines show a faster reduction. At 50 cm depth, the data base at both chronosequences is not so extensive. Whereas the conductivity curves at the S-PM forefield are very heterogeneous, the curves at the C-PM forefield are very close to each other with only a few single curves from the

two oldest moraine showing a steeper slope towards increasing soil matric potential. At the S-PM forefield, the curves show a strong reduction in conductivities at higher matric potential, but this appears to occur at even higher matric potentials at the oldest moraine.

## 5   Data quality and uncertainties

While sampling, transport and analyses of the samples were handled with the uttermost care, we cannot entirely exclude the

possible occurrence of different adverse effects which would negatively impact our measurement uncertainty. A quantification of these uncertainties is difficult to achieve, however, we will briefly discuss them below and give some indication for which of the samples uncertainties might be higher than for others.

All data shown were obtained on the basis of soil samples taken in the field. Soil samples are subject to various types of uncertainties. Due to the high spatial variability in vegetation cover and soil properties, it is difficult to represent a complete

moraine by taking individual soil samples. In order to counteract this situation, we used the variability in vegetation cover as a proxy to bracket this variability. Thus, samples were taken at several locations with different vegetation complexity (low, medium, high) and by taking at least two replicates.

Undisturbed sampling with sample rings that are hammered into the ground is difficult to guarantee and requires great care. In alpine locations, the high stone content of the soil leads to an increased level of difficulty. This was particularly the case

at the young moraines. Sampling was particularly difficult here and required repeated attempts to obtain a sample (which is likely biased towards less or smaller stones). Furthermore, the samples must be transported with as little disturbance as possible. Despite adequate precautions, it cannot be ruled out that vibrations occurred during the transport of the samples, which could have affected the structure of the soil samples. This applies particularly to the C-PM site, where the samples had





to be transported part of the way by helicopter due to the difficult access to the site.

Even when processing the samples in the laboratory, handling of the samples can affect the structure. Also, the integrity of the samples can be influenced by laboratory methods. The complete saturation of the soil samples, which is necessary for many analyses but is rarely found under natural conditions, can lead to the displacement of fine particles in the sample. This could

affect the results of the soil hydraulic properties analysis. However, the resulting uncertainty cannot be quantified. Additionally, the particle size analysis of the samples at a depth of 10 cm in the 10 000-year-old moraine could only be carried out on two samples at the S-PM site since the organic matter content of the other samples was too high. The results of the two samples are subject to strong uncertainties since it could not be ensured that the entire organic matter could be removed. This also accounts for the other soil samples with a high organic matter content. The incomplete removal of the organic matter can lead to an

overestimation of the silt and clay content.

Uncertainties in the analysis of the soil hydraulic properties result, on the one hand, from the already mentioned possible particle displacement during saturation or from the measuring device used. The installation of the tensiometers can lead to a compaction of the soil material around the tensiometer tips, which is more likely in the samples of the old moraines with strongly developed soil material. In the samples of the young moraines, on the other hand, the installation can shift stones and

soil material, which affects the structure. When evaluating the evaporation experiment, the validity of the assumptions about the matric potential distribution must also be considered. Since a drifting apart of the matric potential in the upper and lower tensiomter excludes an approximately constant linear profile of the matric potential in the center of the sample, an evaluation of the conductivity curve is no longer justifiable. When evaluating the retention curves, this was not considered to be critical, since the averaged measured matric potential is related to the bulk water content, determined via the weight measurement.

## 6 Summary

The evolution of soil physical and hydraulic properties over ten millenia was investigated by analyzing soil samples of soil chronosequences in two glacier forefields. One glacier forefield developed on silicate and the other on calcareous parent material. The chosen soil chronosequences consisted of four age classes ranging in age from less than 100 years to more than 10 000 years. Spatial variability in soil properties was taken into account by selecting three sampling sites per moraine

along a gradient of vegetation complexity (low, medium, high) and by taking replicate samples. Uncertainties due to sampling techniques, transportation or sample handling during laboratory analyses cannot be excluded, but were minimized as much as possible through careful handling. Soil physical properties in form of bulk density, porosity, gravel-, sand-, silt-, and clay fraction, as well as organic matter content were analyzed in 10, 30, and 50 cm soil depth.

At both chronosequences a decrease in bulk density and an increase in porosity was observed in all depths. The trend is equally

present at both chronosequences, but absolute values and depth profiles differ among the two parent materials at both sites. The grain size distribution shows a pronounced reduction in sand fraction and an increase in silt and clay fraction over time. The sand fraction at the calcareous site was initially lower than at the silicate site. This difference became stronger at the two oldest age classes, since the reduction in sand fraction was more pronounced at the calcareous site. The silt fraction, however, was



almost equal at the youngest age classes at both sites, but increased strongly in the topsoil of the calcareous site. The smallest changes occurred in the clay fraction, which was higher at the calcareous site at all age classes.

The organic matter content also increased with increasing age at both sites. Whereas the organic matter content in the topsoil was higher at the calcareous site at the intermediate age classes, the silicate site showed a strong increase and a high variation

at the oldest age class. With the change in physical soil properties and organic matter content also a pronounced change in hydraulic soil properties in form of retention curve and hydraulic conductivity curve was observed. The changes reveal an evolution from fast draining coarse textured soils to slow draining soils with high water holding capacity. With the increase in water holding capacity being more pronounced in the top soil at the silicate site. This also affected the evolution of hydrologic flow paths at these sites as shown in Hartmann et al. (2020a).

The obtained data set provides important insights into the development and dynamics of soil structure and soil hydraulic properties for two different parent materials and thus can be useful for the understanding of interactions of pedogenic, biotic, geomorphic and hydrologic processes during landscape evolution.

*Data availability.* The data is available at the online repository of the German Research Center for Geosciences (GFZ, Hartmann et al. (2020b)) and can be accessed under the temporary link:

http://pmd.gfz-potsdam.de/panmetaworks/review/f46bd4d822a0766a9c0baf356bc7e55644d65d62d7ab71527f5d80c35eed11e5

The data will be published with the DOI specified under the link.

*Acknowledgements.* This work was funded by the German Research Foundation (DFG) and the Swiss National Science Foundation (SNF) within the DFG-SNF-project Hillscape (Hillslope Chronosequence and Process Evolution). We thank Jonas Freymüller, Nina Zahn, Wibke Richter, Louisa Kanis, Peter Grosse and Carlo Seehaus for their persevering assistance in the field and Franziska Röpke for her patient

assistance in the laboratory. We also thank Kraftwerke Oberhasli AG (KWO) for permission to conduct the fieldwork at the Stone Glacier forefield and the Canton Uri, the community Unterschächen and Korporation Uri for permission to conduct the fieldwork at the Griessfirn forefield. Many thanks to Thomas Michel and his team of the Alpin Center Sustenpass and Peter Luchs as well as Christine, Franz and Matthias Stadler at Chammlialp for their support and kind hospitality.





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
