# Peer review of "The impact of landscape evolution on soil physics: Evolution of soil physical and hydraulic properties along two chronosequences of proglacial moraines"

_Earth System Science Data, 2020_

## Referee Comment (RC1) · Anonymous Referee #1 · 12 Aug 2020

The manuscript presents experimental data of soil physical and hydraulic properties along glacial moraines of different ages. These basic soil properties (texture, bulk density, porosity, organic carbon content, water retention and hydraulic conductivity) provided here are very useful because they are essential for any quantitative modelling of water and element balances of such soil ecosystems. I congratulate the authors for such a large effort and service for the scientific community. The data basically confirm the theoretically expected pedologic and soil structural development; however, since similar data in comparable quality are extremely rare and relatively difficult to obtain

in reproducible way, so much the better are those of the current manuscript. I would also not be too disappointed about the problems with the description of the hydraulic conductivity functions based on the evaporation data. These data if made available maybe analysed in the future with other methods; it did not seem to me that it was the aim to do it in this data paper. Still, I have a few comments and suggestions for clarification, discussion, and possible improvements:

1. The soil depth is defined related to the current soil surface. During the long times of development, the surface topography may have changed (erosion, colluviation) such that the surface-depth relation could be different at the different locations. This may affect the variability in space and time. Would it be possible, perhaps for future studies, to identify an alternative reference such as, for example, the depth to the intact parent material or other marker? 2. I missed a soil profile description or classification – even a simplified description of soil type and soil horizon characterization would increase the information content on conditions in the sampled soil depths. 3. The particle-size analysis seems to be non-standard, so this could be described a bit more detailed. The sample preparation and dryness state (air, oven) of "Dry sieving", for example, could be defined, the samples should then not be aggregated in any form. Usually, the organic matter and the carbonates are destroyed before wet sieving, and dispersion agent is added. Of course, for the carbonaceous parent material, another method is needed and also the methods to distinguish between organic and inorganic carbon content complicate the analyses. The organic particles could also be water repellent. 4. The discussion (Page 19) on problems with the evaporation method seems too detailed in comparison to other aspects; it shifts the focus too much towards critical evaluation of the application of this method. 5. Overall, the text could be condensed a bit. 6. The use of the past tense and the present tense in the English text is not always consistent and should be checked.

---

## Referee Comment (RC2) · Anonymous Referee #2 · 6 Sep 2020

The manuscript addresses relevant scientific questions within the scope of ESSD. The findings correspond to the previous researches on soil chronosequences, but go further including new, not studied well before, physical and hydraulic properties in the analysis. The manuscript well written, scientific methods and assumptions are valid and clearly outlined. I recommend the manuscript for publication after some minor revisions.

1. The introduction gives the general view on the previous soil chronosequences studies, however the number of such studies is so large that it is not clear why the authors have chosen the papers they have referred. Please give a few words to explain the

choice.

2. The characteristics of the objects are very superficial – no topography and slope characteristics, but according to the Figure 1 they are important in understanding of soil features of the chosen chronosequences. The authors give the reference for the vegetation distribution but it is not characterized in the paper at all. The main criticism is related to the absence of the names of soil types under study. The description of soils is also absent. I recommend including the full characteristics of study objects in the supplementary data but soil type names should be included in the main text.

3. The study objects are not CHRONOSEQUENCES but TOPOCHRONOSE-QUENCES with essential difference of topographic locations of different ages. For example, the soil of 110 yrs will never have such characteristics as the soil of 13,5 kys, as its drainage conditions are initially different. Surely, it is almost impossible to find an ideal chronosequence, especially in mountainous conditions. However, it is worth to explain it clear in the Discussion.

---

## Author Comment (AC2) · 10 Sep 2020

**Response to Reviewer comments**

**Response to Reviewer 2**

**General Comments**

The manuscript addresses relevant scientific questions within the scope of ESSD. The findings correspond to the previous researches on soil chronosequences, but go further including new, not studied well before, physical and hydraulic properties in the analysis. The manuscript well written, scientific methods and assumptions are valid and clearly outlined. I recommend the manuscript for publication after some minor revisions.

**Response to General Comments**

The authors would like to thank the reviewer for spending his/her time on this review and for making valuable comments to improve our manuscript. We will address these comments and suggestions below.

1. The introduction gives the general view on the previous soil chronosequences studies, however the number of such studies is so large that it is not clear why the authors have chosen the papers they have referred. Please give a few words to explain the choice.

We agree that there is a large number of soil chronosequence studies. The chosen papers specifically deal with the investigation of soil physical and chemical changes along soil chronosequences. We thus give an exemplary overview of studies which are most comparable to our presented study. A full literature review is beyond the scope of our data presentation. We will add a sentence to clarify this.

2. The characteristics of the objects are very superficial – no topography and slope characteristics, but according to the Figure 1 they are important in understanding of soil features of the chosen chronosequences. The authors give the reference for the vegetation distribution but it is not characterized in the paper at all. The main criticism is related to the absence of the names of soil types under study. The description of soils is also absent. I recommend including the full characteristics of study objects in the supplementary data but soil type names should be included in the main text.

We agree that some more information on topography, slope characteristic, soil type and soil horizon characterization would be helpful and we will include this in the revised manuscript. We will include a table containing information on elevation, slope, dominant vegetation, vegetation cover and soil type and update the site descriptions:

We will include the following information in section 2.1.1 (Silicate parent material):

*Page 5 Line 4: "Table 1 provides an overview of the main characteristics of the 4 moraines including elevation, slope, dominant vegetation, vegetation cover, and soil type [Maier et al., 2019, Musso et al. 2020]. The soil at the two youngest moraines was classified as a Hyperskelectic Leptosol. At the 3 000 years old moraine a Skelectic Cambisol and at the oldest moraine an Entic Podzol was found. Illustrations of the horizontal soil layers at each moraine can be found in Maier et al. (2019). The vegetation cover differs significantly among the four age classes and was mapped in summer 2017 (Maier et al., 2019). The moraines are occasionally grazed by cows and sheep during the summer months, which we prevented during our study by the installation of fences. Whereas the vegetation cover at the oldest moraine was dominated by a variety of prostrate shrubs, small trees and several grasses, the 3 000-year-*

*old moraine has mainly a grassland cover with fern, mosses, sedges and forbs. The 160-year-old moraine was dominated by grasses, lichen, forbs, and shrubs. The vegetation cover of the youngest moraine was sparse with mainly grass, moss, forbs, and a few shrubs."*

We will update the text in section 2.1.2 (Calcareous parent material) line 18-22 to:

*"The four selected moraines were dated by Musso et al. (2019) based on historical maps and the radiocarbon method. The youngest moraine is 110 years old and is located at 2200 m a.s.l. The three other moraines are 160, 4 900, and 13 500 years old and located at an elevation of roughly 2030 m a.s.l. (see Fig. 1). An overview of the main characteristics of the 4 moraines including elevation, slope, dominant vegetation, vegetation cover, and soil type [Maier et al., 2019, Musso et al., 2019, Musso et al., 2020] is provided in Table 1. The soil at the two youngest moraines was classified as a Hyperskeletic Leptosol and at the two oldest moraines as a Calcaric Skeletic Cambisol (Musso et al., 2019). The two oldest moraines were densely covered with grass, dwarf shrubs and sedge. The vegetation coverage of the two younger moraines was sparse with patches of grass and forbs at the 160-year-old moraine and patches of mostly mosses and lichens at the 110-year-old moraine."*

3. The study objects are not CHRONOSEQUENCES but TOPOCHRONOSEQUENCES with essential difference of topographic locations of different ages. For example, the soil of 110 yrs will never have such characteristics as the soil of 13,5 kys, as its drainage conditions are initially different. Surely, it is almost impossible to find an ideal chronosequence, especially in mountainous conditions. However, it is worth to explain it clear in the Discussion.

We agree on the need to discuss the limitation of the chronosequence approach and will include the following into the text:

*"The space for time substitution approach assumes that a sequence of sites (e.g. moraines) with similar site characteristics like topography, climate and parent material can be treated as a chronosequence.
It is well known that the application of this chronosequence concept has some limitations.
The assumption that time is the only factor affecting soil development in a spatial sequence of soils is rarely valid, but the only option for a detailed historical reconstruction of the soil development at a particular location (Phillips 2015). We therefore have to assume that differences in topography and elevation among the selected moraines only lead to moderate differences in soil hydrologic conditions. However, we made sure that slopes of the three selected plots per moraine were in a similar range. The plots at the silicate parent material range in slope from 18 to 34° with the majority of plots between 20 and 30°. The maximum elevation difference between the lowest and the highest plot is 108 m.
At the calcareous site the slopes range from 27 to 44°, also with the majority of plots ranging between 20 and 30°. Here, three out of four moraines are at almost the same elevation. The elevation difference to the youngest moraine is 170 m."*

---

## Author Response (AR1)

Dear Editor,

We would like to thank you for giving us the opportunity to revise our manuscript. We have revised the manuscript entitled "The impact of landscape evolution on soil physics: Evolution of soil physical and hydraulic properties along two chronosequences of proglacial moraines" in response to the reviewers' comments. Please find attached a revised version of this manuscript as well as a detailed list of our responses to these comments.

We are grateful to you and the reviewers for your interest in our paper and for the detailed evaluation, valuable suggestions, and recommendations. As you will see when examining our revision, the reviewers' comments and recommendations were taken seriously and were thoroughly addressed in our revised paper.

**Response to Reviewer comments**

**Response to Reviewer 1**

**General Comments**
The manuscript presents experimental data of soil physical and hydraulic properties along glacial moraines of different ages. These basic soil properties (texture, bulk density, porosity, organic carbon content, water retention and hydraulic conductivity) provided here are very useful because they are essential for any quantitative modelling of water and element balances of such soil ecosystems. I congratulate the authors for such a large effort and service for the scientific community. The data basically confirm the theoretically expected pedologic and soil structural development; however, since similar data in comparable quality are extremely rare and relatively difficult to obtain in reproducible way, so much the better are those of the current manuscript. I would also not be too disappointed about the problems with the description of the hydraulic conductivity functions based on the evaporation data. These data if made available maybe analysed in the future with other methods; it did not seem to me that it was the aim to do it in this data paper. Still, I have a few comments and suggestions for clarification, discussion, and possible improvements:

**Response to General Comments**
The authors would like to thank the reviewer for spending his/her time on this review and for making valuable comments to improve our manuscript. We are happy to see that our efforts and the data set are appreciated! We will address all comments and suggestions below.

1. The soil depth is defined related to the current soil surface. During the long times of development, the surface topography may have changed (erosion, colluviation) such that the surface-depth relation could be different at the different locations. This may affect the variability in space and time. Would it be possible, perhaps for future studies, to identify an alternative reference such as, for example, the depth to the intact parent material or other marker?

We agree with the uncertainties caused by defining the soil surface as the reference point. We agree that for example the depth to the intact parent material could be an alternate marker that excludes those uncertainties. However, locating (and also reaching) the depth of intact parent material can often be difficult.

2. I missed a soil profile description or classification – even a simplified description of soil type and soil horizon characterization would increase the information content on conditions in the sampled soil depths.

We agree that some more information on soil type and soil horizon characterization would be helpful and have added this in the revised manuscript. We also included some more information on elevation, slope, dominant vegetation, vegetation cover and soil type in form of a table. We also updated the site descriptions a follows:

We included the following information in section 2.1.1 (Silicate parent material):

Page 5 Line 4: *"Table 1 provides an overview of the main characteristics of the 4 moraines including elevation, slope, dominant vegetation, vegetation cover, and soil type [Maier et al., 2019, Musso et al. 2020]. The soil at the two youngest moraines was classified as a Hyperskelectic Leptosol. At the 3 000 years old moraine a Skelectic Cambisol and at the oldest moraine an Entic Podzol was found. Illustrations of the soil layers at each moraine can be found in Maier et al. (2019). The vegetation cover differs significantly among the four age classes and was mapped in summer 2017 (Maier et al., 2019). The moraines are occasionally grazed by cows and sheep during the summer months, which we prevented during our study by the installation of fences. Whereas the vegetation cover at the oldest moraine was dominated by a variety of prostrate shrubs, small trees and several grasses, the 3 000-year-old moraine has mainly a grassland cover with fern, mosses, sedges and forbs. The 160-year-old moraine was dominated by grasses, lichen, forbs, and shrubs. The vegetation cover of the youngest moraine was sparse with mainly grass, moss, forbs, and a few shrubs."*

We updated the text in section 2.1.2 (Calcareous parent material), page 6, line 23-30 to:

[revised manuscript text omitted]

3. The particle-size analysis seems to be non-standard, so this could be described a bit more detailed. The sample preparation and dryness state (air, oven) of "Dry sieving", for example, could be defined, the samples should then not be aggregated in any form. Usually, the organic matter and the carbonates are destroyed before wet sieving, and dispersion agent is added. Of course, for the carbonaceous parent material, another method is needed and also the methods to distinguish between organic and inorganic carbon content complicate the analyses. The organic particles could also be water repellent.

We agree, that the description of the method used for the particle size analysis could be described in more detail. We updated the manuscript accordingly on page 7, line 24 – page 8, line 9 as follows:

*"For the grain size analysis, we used a combination of dry sieving (particles > 0.063 mm) and sedimentation analysis (particles < 0.063 mm) with the hydrometer method. Particles between 2 mm and 0.063 mm were classifiesd as sand, between 0.063 mm and 0.002 mm as silt and < 0.002 mm as clay. Particles larger 0.063 mm were separated from the fine particles by wet sieving. They were then dried at 550 °C for combustion of organic matter prior to the dry sieving. Due to lab limitations organic matter removal from the fine particles was only possible by floating off the lighter fractions prior to the sedimentation analysis. 24 hours before sedimentation analysis, $Na_4P_2O_7$ was added as a dispersant to the sample solution to prevent coagulation of the particles. Particle size fractions were calculated as weight percentages of the fine earth (< 2 mm), thus excluding gravel and stones to avoid that single larger stones shift or dominate the distribution."*

4. The discussion (Page 19) on problems with the evaporation method seems too detailed in comparison to other aspects; it shifts the focus too much towards critical evaluation of the application of this method.

We agree and shortened the text on page 19, lines 1-7 to:

*"Out of the 60 soil samples taken at each moraine and analyzed by the method according to Schindler (1980), only 32 at S-PM and 41 at C-PM could be analyzed for the hydraulic conductivity curve according to the method by Peters and Durner (2008). Seven of the excluded soil samples at S-PM and two at C-PM could not be used because the installation of the second tensiometer into the sample was prevented by stones. The other samples had to be excluded as the upper part of the sample dried out much faster than the lower part during the evaporation experiment. An approximately linear profile of the soil matric potential within the sample could thus no longer be assumed. This is typical for soils with large pore sizes, such as sandy soils."*

5. Overall, the text could be condensed a bit.

We reread the text thoroughly and shortened a few sections. However, we find that a certain level of detail is necessary.

6. The use of the past tense and the present tense in the English text is not always consistent and should be checked.

We agree and corrected the tense.

**Response to Reviewer comments**

**Response to Reviewer 2**

**General Comments**
The manuscript addresses relevant scientific questions within the scope of ESSD. The findings correspond to the previous researches on soil chronosequences, but go further including new, not studied well before, physical and hydraulic properties in the analysis. The manuscript well written, scientific methods and assumptions are valid and clearly outlined. I recommend the manuscript for publication after some minor revisions.

**Response to General Comments**
The authors would like to thank the reviewer for spending his/her time on this review and for making valuable comments to improve our manuscript. We will address these comments and suggestions below.

1. The introduction gives the general view on the previous soil chronosequences studies, however the number of such studies is so large that it is not clear why the authors have chosen the papers they have referred. Please give a few words to explain the choice.

We agree that there is a large number of soil chronosequence studies. The chosen papers specifically deal with the investigation of soil physical and chemical changes along soil chronosequences. We thus give an exemplary overview of studies which are most comparable to our presented study. A full literature review is beyond the scope of our data presentation. We added this explanation on page 2, line 32 to 34:

*"The listed soil chronosequence studies are not meant to be comprehensive but were selected as they are most comparable to the study presented here. A full literature review on chronosequence studies is beyond the scope of our data presentation."*

2. The characteristics of the objects are very superficial – no topography and slope characteristics, but according to the Figure 1 they are important in understanding of soil features of the chosen chronosequences. The authors give the reference for the vegetation distribution but it is not characterized in the paper at all. The main criticism is related to the absence of the names of soil types under study. The description of soils is also absent. I recommend including the full characteristics of study objects in the supplementary data but soil type names should be included in the main text.

We agree that some more information on slope, soil type and soil horizon characterization would be helpful. We included a table containing information on elevation, slope, dominant vegetation, vegetation cover and soil type and updated the site descriptions:

We included the following information in section 2.1.1 (Silicate parent material):

Page 5 Line 4: *"Table 1 provides an overview of the main characteristics of the 4 moraines including elevation, slope, dominant vegetation, vegetation cover, and soil type [Maier et al., 2019, Musso et al. 2020]. The soil at the two youngest moraines was classified as a Hyperskelectic Leptosol. At the 3 000 years old moraine a Skelectic Cambisol and at the oldest moraine an Entic Podzol was found. Illustrations of the horizontal soil layers at each moraine can be found in Maier et al. (2019). The vegetation cover differs significantly among the four age classes and was mapped in summer 2017 (Maier et al., 2019).*

*The moraines are occasionally grazed by cows and sheep during the summer months, which we prevented during our study by the installation of fences. Whereas the vegetation cover at the oldest moraine was dominated by a variety of prostrate shrubs, small trees and several grasses, the 3 000-year-old moraine has mainly a grassland cover with fern, mosses, sedges and forbs. The 160-year-old moraine was dominated by grasses, lichen, forbs, and shrubs. The vegetation cover of the youngest moraine was sparse with mainly grass, moss, forbs, and a few shrubs."*

We updated the text in section 2.1.2 (Calcareous parent material), page 6, line 23-30 to:

*"The four selected moraines were dated by Musso et al. (2019) based on historical maps and the radiocarbon method. The youngest moraine is 110 years old and is located at 2200 m a.s.l. The three other moraines are 160, 4 900, and 13 500 years old and located at an elevation of roughly 2030 m a.s.l. (see Fig. 1). An overview of the main characteristics of the 4 moraines including elevation, slope, dominant vegetation, vegetation cover, and soil type [Maier et al., 2019, Musso et al., 2019, Musso et al., 2020] is provided in Table 1. The soil at the two youngest moraines was classified as a Hyperskeletic Leptosol and at the two oldest moraines as a Calcaric Skeletic Cambisol (Musso et al., 2019). The two oldest moraines were densely covered with grass, dwarf shrubs and sedge. The vegetation coverage of the two younger moraines was sparse with patches of grass and forbs at the 160-year-old moraine and patches of mostly mosses and lichens at the 110-year-old moraine."*

**Table 1.** Overview of the main characteristics of the four moraines at the silicate and calcareous parent material. This information was compiled from the publications Maier et al. (2020), Musso et al. (2019), and Musso et al. (2020).

| | Moraine age [years] | Complexity level | Elevation [m.a.s.l.] | Slope [°] | Aspect | Dominant vegetation | Vegetation Cover [%] | Soil type |
|---|---|---|---|---|---|---|---|---|
| *Silicate parent material* | | | | | | | | |
| | 30 | low | 1952 | 21 | NE | Salix hastata | 50 | Hyperskeletic Leptosol |
| | 30 | medium | 1959 | 34 | NE | Epilobium fleischeri, Poa alpina | 30 | Hyperskeletic Leptosol |
| | 30 | high | 1955 | 23 | NE | Salix retusa, Trifolium pallescens | 45 | Hyperskeletic Leptosol |
| | 160 | low | 1989 | 25 | NE | Anthoxanthum alpinum, Salix retusa | 80 | Hyperskeletic Leptosol |
| | 160 | medium | 1981 | 31 | NE | Campanula scheuchzeri, Trifolium pallescens | 80 | Hyperskeletic Leptosol |
| | 160 | high | 1989 | 26 | NE | Salix glaucosericea, Anthoxanthum alpinum | 95 | Hyperskeletic Leptosol |
| | 3 000 | low | 1914 | 32 | S | Carlina acaulis, Achillea moschata | 60 | Skeletic Cambisol |
| | 3 000 | medium | 1910 | 32 | S | Vaccinium vitis-idaea, Carlina acaulis | 85 | Skeletic Cambisol |
| | 3 000 | high | 1888 | 25 | SE | Thymus polytrichus, Trifolium nivale | 70 | Skeletic Cambisol |
| | 10 000 | low | 1882 | 24 | NE | Rhododendron ferrugineum, Vaccinium myrtillus | 100 | Dystric Cambisol |
| | 10 000 | medium | 1882 | 29 | N | Rhododendron ferrugineum, Vaccinium uliginosum | 90 | Dystric Cambisol |
| | 10 000 | high | 1873 | 18 | NE | Rhododendron ferrugineum, Calluna vulgaris | 90 | Dystric Cambisol |
| *Calcareous parent material* | | | | | | | | |
| | 110 | low | ~2200 | - | WNW | Saxifraga aizoides, Poa alpina | 50 | Hyperskeletic Leptosol |
| | 110 | medium | ~2200 | - | WNW | Saxifraga aizoides, Poa alpina | 52 | Hyperskeletic Leptosol |
| | 110 | high | ~2200 | - | WNW | axifraga aizoides, Poa alpina | 63 | Hyperskeletic Leptosol |
| | 160 | low | 2038 | 35 | E | Dryas octopetala, Saxifraga aizoides | 70 | Hyperskeletic Leptosol |
| | 160 | medium | 2025 | 33 | NE | Astragalus alpinus, Dryas octopetala | 78 | Hyperskeletic Leptosol |
| | 160 | high | 2032 | 29 | NW | Salix retusa, Festuca quadriflora | 79 | Hyperskeletic Leptosol |
| | 4 900 | low | 2019 | 28 | SE | Anthyllis vulneraria, Lotus alpinus | 100 | Calcaric Skeletic Cambisol |
| | 4 900 | medium | 2016 | 33 | NE | Luzula sylvatica ssp sieberi, Lotus alpinus | 100 | Calcaric Skeletic Cambisol |
| | 4 900 | high | 2018 | 34 | W | Leontodon helveticus, Festuca rubra | 100 | Calcaric Skeletic Cambisol |
| | 13 500 | low | 2001 | 35 | NW | Alchemilla fissa, Ligusticum mutellina, | 100 | Calcaric Skeletic Cambisol |
| | 13 500 | medium | 2012 | 38 | NE | Anthoxanthum alpinum, Dryas octopetala | 100 | Calcaric Skeletic Cambisol |
| | 13 500 | high | 2017 | 33 | NE | Alchemilla conjuncta, Dryas octopetala | 100 | Calcaric Skeletic Cambisol |

3. The study objects are not CHRONOSEQUENCES but TOPOCHRONOSEQUENCES with essential difference of topographic locations of different ages. For example, the soil of 110 yrs will never have such characteristics as the soil of 13,5 kys, as its drainage conditions are initially different. Surely, it is almost impossible to find an ideal chronosequence, especially in mountainous conditions. However, it is worth to explain it clear in the Discussion.

We agree on the need to discuss the limitation of the chronosequence approach and included the following text on page 20, line 19-28:

[revised manuscript text omitted]